# A new point mutation in the HC-Pro of potato virus Y is involved in tobacco vein necrosis

Giuseppe Parrella[1]*, Benoit Moury[2]*

**1** Institute for Sustainable Plant Protection of The National Research Council (IPSP-CNR), Portici, Italy,
**2** INRAE, Pathologie Végétale, Montfavet, France

* giuseppe.parrella@ipsp.cnr.it (GP); benoit.moury@inrae.fr (BM)

## Abstract

Tobacco vein necrosis (TVN) is a complex phenomenon regulated by different genetic determinants mapped in the HC-Pro protein (amino acids $N_{330}$, $K_{391}$ and $E_{410}$) and in two regions of potato virus Y (PVY) genome, corresponding to the cytoplasmic inclusion (CI) protein and the nuclear inclusion protein a-protease (NIa-Pro), respectively. A new determinant of TVN was discovered in the MK isolate of PVY which, although carried the HC-Pro determinants associated to TVN, did not induce TVN. The HC-Pro open reading frame (ORF) of the necrotic infectious clone PVY N605 was replaced with that of the non-necrotic MK isolate, which differed only by one amino acid at position 392 ($T_{392}$ instead of $I_{392}$). The cDNA clone N605_MKHCPro inoculated in tobacco induced only weak mosaics at the systemic level, demostrating that the amino acid at position 392 is a new determinant for TVN. No significant difference in accumulation in tobacco was observed between N605 and N605_MKHCPro. Since phylogenetic analyses showed that the loss of necrosis in tobacco has occurred several times independently during PVY evolution, these repeated evolutions strongly suggest that tobacco necrosis is a costly trait in PVY.

**Data Availability Statement:** All relevant data are within the paper and its Supporting Information files.

**Funding:** This research was partially funded by the Italian National Research Council (CNR) with the

## Introduction

Potato virus Y (PVY) is one of the most important plant viruses with a worldwide distribution [1] and infects a wide range of host plants comprising 495 species in 72 genera from 31 families, including numerous economically important solanaceous crop plants, *e.g.* potato (*Solanum tuberosum*), pepper (*Capsicum* spp.), tomato (*S. lycopersicum*) and tobacco (*Nicotiana tabacum*) [2].

PVY belongs to the genus *Potyvirus* in the family *Potyviridae*. It has flexuous and filamentous virions with a length of about 730–740 nm and a diameter of 11 nm. The 5'-end of the genome is covalently linked to a genome-linked viral protein (VPg) through a tyrosine residue and the 3'-end contains a poly(A)-tail. As in other potyviruses, its genome consist of a positive-sense and single-stranded RNA (ssRNA) molecule of about 9.7 kb that is translated into a single polyprotein of about 3062 amino acid (aa) residues. It is transmitted at least by 70 species of aphids in a non-persistent manner [3].

Short Term Mobility (STM) program 2014. The funders had no role in study design, data collection and analysis, decision to publish, or preparation of the manuscript.

**Competing interests:** The authors have declared that no competing interests exist.

PVY exists in a complex of strain groups and variants. PVY classification is based on: i) serology–determined by antibody recognition of the CP, ii) biological properties, like symptomatology induced in certain hosts (e.g. tobacco) and hypersensitive response (HR) in different potato cultivars carrying resistance genes and iii) phylogenetic relationships among isolates based on genome sequence analyses, including number and positions of recombination junctions (RJs). Serological, molecular and biological properties of PVY strains are not always correlating, because there are numerous PVY strains/variants that have similar serological classification and biological properties but different molecular properties, such as the strains inducing tobacco veinal necrosis (TVN) in *N. tabacum* cvs. Xanthi and Burley: $PVY^N$, $PVY^{NTN}$, $PVY^{NA-NTN}$, $PVY^{N:O}$, $PVY^{N:O-B}$, $PVY^{N-W}$ and $PVY^{NTN-NW}$ [4,5]. Several molecular determinants involved in TVN have been mapped on the PVY helper component-proteinase (HC-Pro). Based on a pioneer work by Glais et al. [6], who identified the putative TVN domain between the 3' end of the P1 gene and the 5' end of the P3 gene, by reverse genetics approach, Tribodet et al. [7] demonstrated that amino acids $K_{391}$ and $E_{410}$ in the C-terminal part of the HC-Pro were directly involved in the TVN symptoms. Nevertheless, as reported for the L26 strain of PVY, the genetic determinant of TVN is more complex and involves other amino acid substitutions in the C-terminal part of the HC-Pro in addition to $K_{391}$ and $E_{410}$ [8]. In the following, numbering of amino acid positions in the HC-Pro follows Adams et al. [9] who determined a new position for the P1/HC-Pro cleavage site of PVY, which was 9 residues downstream from the original position used for the numbering in Tribodet et al. [7] and subsequent articles [10].

A PVY isolate, MK, collected in Italy in 2009 from a *Datura metel* plant, showed unusual biological and serological characteristics. It was identified serologically as a N or NTN type of isolate. Nevertheless, symptoms induced in tobacco consisted in vein clearing and not in vein necrosis as expected.

The objectives of the present work are therefore the complete characterization of the MK isolate and the identification of the mutation(s) responsible for its unusual characteristics.

## Materials and methods

### PVY isolates and infectious clone

The MK isolate of PVY was collected in summer 2009 from a *Datura metel* plant showing green mosaic (Fig 1), which grew spontaneously in the area surrounding a small farm, located in the municipality of Alberobello (province of Bari, southern Italy). The infected plant was collected, transplanted in a pot of 20 cm in diameter, maintained in an insect-proof greenhouse and checked serologically against different viruses (cucumber mosaic virus, tomato spotted wilt virus, parietaria mottle virus, potato leaf roll virus, potato virus X, potato virus M, potato virus S and PVY). Only PVY was detected. Moreover, electron microscope observations of *D. metel* sap extracts by dip method [11], revealed only the presence of elongated virus particles and laminates typical of a potyvirus infection [12].

Isolate PVY-MK was maintained only in *D. metel* plants and checked periodically by RT-PCR, followed by Sanger sequencing of amplicons corresponding to the HCPro open reading frame (ORF), and ELISA with a panel of antisera commercially available. Original inoculum was also stored at 5˚C as dehydrated leaves, under continuous exposure to calcium chloride.

Virus obtained from the N605 infectious clone [13] was used as a reference for the $PVY^N$ group. In addition, isolates Sot-1 and Nit-1, belonging to $PVY^{NTN}$ and $PVY^{Ob}$ groups, respectively [14,15], were used as controls for the biological and serological assays.

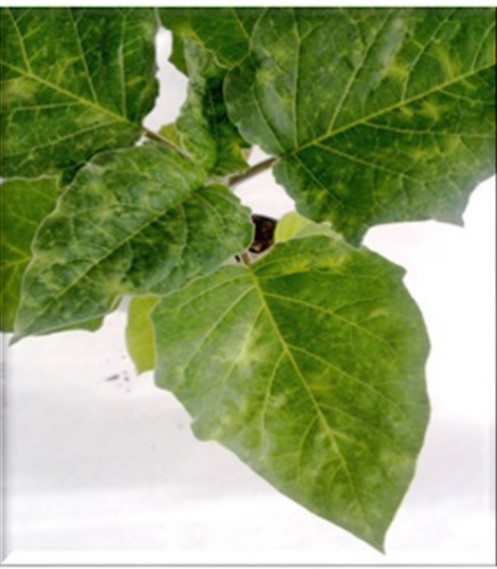

**Fig 1. Original *Datura metel* plant showing mosaic associated to PVY MK infection.**

## Biological and serological characterization of PVY-MK

A set of plant species including *Chenopodium amaranticolor*, *C. quinoa*, *D. metel*, *N. tabacum* cvs. Samsun and Xanthi, *N. glutinosa*, *N. benthamiana*, *N. clevelandii* and *S. lycopersicum* 'Momor'was used to characterize the pathotype of the MK isolate. These plants (two replicates per species or accession) were mechanically inoculated at the 4–6 leaf stage with sap from PVY-MK-infected *D. metel* plants and with those obtained from tobacco plants infected by Sot-1 and Nit-1, following Parrella, [16]. Inoculated plants were maintained in an insect-proof greenhouse at 24/18°C day/night, under natural light conditions. Every week, they were fertilized by adding complete nutritive complex to the irrigation water. Symptoms were recorded weekly up to 2 months after inoculation.

Inoculated plants and control healthy plants were tested by DAS-ELISA 15 and 30 days post inoculation (dpi), essentially as described by Clark and Adams [17], against one polyclonal and two monoclonal antibodies: the "PVY polyclonal", recognizing all the PVY serotypes, the "PVY monoclonal" recognizing $PVY^{Oa}$, $PVY^{C}$, $PVY^{N}$ and $PVY^{NTN}$, and the "$PVY^{N}$ monoclonal" recognizing specifically $PVY^{N}$ and $PVY^{NTN}$ serotypes (Bioreba AG, Switzerland) [15]. The assay was conducted in duplicate for each sample in polystyrene microtitre plates (Nunc$^{TM}$ MicroWell$^{TM}$ 96-Well Microplates, Thermo Scientific$^{TM}$) according to manufacturer's instructions. The mean of absorbance value at 405 nm ($OD^{405}$) was recorded using a Titertek Multiskan photometer (Thermo Scientific$^{TM}$) every 30 min, during 2h of incubation with substrate at room temperature. Samples were considered positive if leaf extract gave an $OD^{405}$ signal at least twice that obtained from the extract of a healthy plant.

## RNA isolation, cloning and sequencing of the full-length PVY-MK genome

Total RNA was isolated from a naturally-infected *D. metel* plant and used to generate the complete genome sequence of PVY-MK by RT-PCR with seven primer pairs defined from the sequence of isolate PVY-12 (Acc. No. AB185833; S1 Table), since this isolate showed serological features similar to PVY-MK [18]. The S, M4 and M4-T primers were used for the amplification of the 5' end of the genome [19], while the 3' end of the genome was amplified using the

SMARTer®RACE 5'/3' kit (Takara Bio Inc. Shiga, Japan), following manufacturer's instructions. Overlapping RT-PCR fragments of approximately 1000 to 1800 bp were cloned into the pGEM-T Easy vector (Promega Corp., Madison, WI, U.S.A.) according to standard protocols [20]. One to three clones per fragment were sequenced (Genome Express, Grenoble, France).

## Sequence analyses

Sequences of the primers used to generate the cloned cDNA fragments were removed before analysis. The overlapping sequence contigs were assembled with Contig Assembly Program [21]. Amino acid sequences were then aligned and compared with the sequence of PVY-12 with the Clustal W program [22], using default parameters on the BioEdit software package [23].

The Maximum-likelihood method was utilized to construct phylogenetic tree of PVY isolates, using 1000 bootstrap replicates and the best nucleotide substitution model selected by MEGA software (version 7.0.26) [24].

The RDP2 software [25] was used to detect recombination events in the PVY-MK genome using a set of reference sequences of the N, O and C groups and default parameters for all recombination methods implemented in the software (RDP, GENECONV, BootScan, Max-Chi, Chimera, SiScan, 3Seq, LARD and Phylpro) [25].

## Construction of the chimeric infectious PVY clone N605_MKHCPro

The infectious cDNA clone of PVY isolate N605 was modified (i) by inserting short plant intron sequences to increase its stability [26,27] and (ii) by including a cassette containing the 2μ yeast replication origin and a selectable marker (Trp-1 promoter and gene) as described in Fernandez-Delmond et al. [28] and Ayme et al. [29], allowing homologous recombination in yeast. The region coding the HC-Pro was deleted by reverse PCR from the N605 clone and a unique *Not*I restriction site was introduced at the junction, yielding the acceptor shuttle vector. The entire cDNA corresponding to the HC-Pro coding sequence of the MK isolate was obtained with the primers SON41HCdeb (5'-TATGATGCACGTTCCAAGGTTACTCAAGGCG; positions 988–1018) and SON41HCend (5'-GCATGCTCCAGGAATACCACCAACTCTATAATG; positions 2427–2395). Yeast was co-transformed with the shuttle vector linearized with *Not*I and with cDNA of the MK HC-Pro coding sequence and yielded through homologous recombination the chimeric clone N605_MKHCPro. The chimeric clone was sequenced in order to check its conformity. Infectivity of the chimeric clone was checked by DNA-coated tungsten particle bombardments of juvenile plants (four leaf stage) of *N. clevelandii* and *N. benthamiana*. The N605 infectious clone was used as control.

## Comparison of accumulation of PVY N605 clone and chimeric clone N605_MKHCPro

The accumulation at the systemic level of N605 and N605_MKHCPro were compared in *N. tabacum* cv. Xanthi. Inocula of both viruses were prepared separately from infected Xanthi plants, calibrated using quantitative DAS-ELISA using curves representing $OD^{405}$ in function of dilution factors as described in Moury and Simon [27] and used for inoculation of 20 Xanthi plants per virus. Inoculated plants were placed in a greenhouse in a completely random design. Quantitative DAS-ELISAs were performed to estimate the relative PVY concentration in and the inoculated Xanthi plants at 15 and 30 dpi (10 plants per virus at each timepoint) and comparisons between the two PVY variants at each date were performed using Kruskal-Wallis tests [30].

## Results

### Biological and serological characterization of isolate MK

The MK isolate induced only vein clearing in tobacco cvs. Xanthi and Samsun, like symptoms reported for non-necrotic PVY isolates, e.g. isolate N it-1 (Table 1; [14]). PVY-MK did not induce TVN, typical of necrotic isolates, up to 60 dpi, while Sot-1 and N605 showed distinct TVN, sometimes associated with leaf distortions, within 14–21 dpi in both Xanthi and Samsun. On other hosts, symptoms were similar among the four tested PVY isolates (Table 1).

In serological tests, PVY-MK, Sot-1 and N605 were clearly detected in infected tobacco plants by "monoclonal PVY", "monoclonal PVY[N]" and "polyclonal PVY" antibodies, while Nit-1 was detected only by "polyclonal PVY" antibody (Table 1).

### Sequence analysis and phylogenetic relationships of MK isolate with PVY reference strains based on the whole genome

The genome of isolate MK was determined to be 9702 nt long, excluding the polyA tail. The sequence has been deposited in the GenBank database under the accession number MF440322. Preliminary BLASTn analysis of the whole genome sequence indicated that the closest match was for strains of the PVY[NTN] group. The PVY isolate closest to MK (98.5% nucleotide identity) was isolate IUNG-4 (accession number JF927752), a tobacco isolate collected in Poland in 2008. Contrary to PVY[NTN] isolates however, MK did not induce veinal necrosis in tobacco after mechanical inoculation. Hence, we suspected that one or a few mutations in MK were responsible for its atypical phenotype on tobacco.

Application of the RDP2 package to the analysis of genome sequence indicated that MK had a typical PVY[NTN] recombination pattern with PVY[O] and PVY[N] identified as its parents. Three recombination breakpoints were confirmed by seven of nine methods (RDP,

**Table 1. Biological and serological properties of PVY isolate MK and reference isolates Sot-1 (PVYNTN), N605 (PVYN) and Nit-1 (PVYO).**

|  | PVY isolate | | | |
|---|---|---|---|---|
|  | **MK** | **Sot-1** | **Nit-1** | **N605** |
| **Biological properties** |  |  |  |  |
| *Chenopodium amaranticolor* | LLc/-[a] | LLc/- | LLc/- | LLc/- |
| *C. quinoa* | LLc/- | LLc/- | LLc/- | LLc/- |
| *Datura metel* | -/mo | -/mo | -/mo | -/mo |
| *Nicotiana tabacum* cv. Xanthi | -/Vc | -/VN, ds | -/Vc | -/VN |
| *N. tabacum* cv. Samsun | -/Vc | -/VN, ds | -/Vc | -/VN |
| *N. benthamiana* | -/mo, ds | -/mo, ds | -/mo, ds | -/mo, ds |
| *N. clevelandii* | -/mo, ds | -/mo, ds | -/mo, ds | -/mo, ds |
| *Solanum lycopersicum* | -/mo | -/mo | -/mo | nt |
| **Serological properties** |  |  |  |  |
| PVY monoclonal[b] | + | + | - | + |
| PVY[N] monoclonal[c] | + | + | - | + |
| PVY polyclonal[d] | + | + | + | + |

[a]local symptoms/ symptoms at the systemic level; LLc = chlorotic local lesions; mo = mosaic; Vc = vein clearing; VN = vein necrosis; ds = distortion; - = no symptoms; nt = not tested.

[b]recognizes PVY[N], PVY[Oa] and PVY[C] [15].

[c]recognizes PVY[N] and PVY[NTN] [15].

[d]recognizes PVY[N], PVY[NTN], PVY[O] and PVY[C].

GENECONV, BootScan, MaxChi, Chimera, SiScan and 3Seq) implemented in RDP2: the first one between nucleotide positions 2385 and 2415 (99% confidence interval - CI) at the HC-Pro/P3 cistron junction, the second one between nucleotide positions 5769 and 5863 (99% CI) in the VPg cistron and the third one between nucleotide positions 8547 and 8598 (99% CI) at the NIb/CP cistron junction. This recombination pattern is highly similar to that of the SYR-I group of PVY [31] (S1 Fig).

The sequence of the HC-Pro cistron of PVY-MK was aligned and compared with those from other reference strains corresponding either to the "N" phylogenetic group (strains N605, L26, PVY-12, SASA61 and LW) or to the "O" phylogenetic group (strain O139) (Table 2). Thirteen nucleotides differed between the sequences of the HC-Pro coding regions of the N605 clone and the MK isolate. The deduced HC-Pro sequences differed only at amino acid position 392 with an isoleucine for the N605 clone and a threonine for isolate MK. Note that there is one amino acid difference between the deduced HC-Pro sequences of the N605 published sequence (GenBank accession number X97895) and the N605 cDNA clone at position 204 (serine for the clone and asparagine for the isolate) (Table 2). Therefore, the amino acid substitution at position 392 is a strong candidate for the lack of necrosis in tobacco of isolate MK.

Blastp analyses (August 5$^{th}$, 2022) with the amino acid sequence of the HC-Pro of isolate MK revealed that (i) more than 100 PVY isolates have a sequence identical to MK except for the $I_{392}T$ substitution and (ii) only one PVY isolate carries the $I_{392}T$ substitution in addition to MK, isolate SASA-61 collected from potato in the UK (accession number AJ585198). Isolate SASA-61 encodes a HC-Pro with both residues $K_{391}$ and $E_{410}$, required for veinal necrosis in tobacco, but does not induce veinal necrosis symptoms on infected tobacco plants, similar to PVY-MK [32,33]. In addition, it belongs to a different group of PVY from that of MK, group NA-N (Fig 2). Consequently, the most parsimonious phylogenetic tree shows that the $I_{392}T$ substitution was acquired at least twice during PVY evolution, once in the MK branch and once in the SASA-61 branch and that it was both times associated to a loss of necrotic symptoms in tobacco. Hence, it is highly probable that this mutation confers a fitness benefit to PVY.

## Symptoms induced at the systemic level in tobacco cultivar Xanthi by PVY N605 clone and chimera N605_MKHCPro

Given the frequent involvement of HC-Pro of PVY as a determinant of symptoms in tobacco [4,7,8], we tested by reverse genetics using the N605 cDNA clone if the HC-Pro cistron of isolate MK was responsible for the lack of veinal necrosis in that host (Fig 3). For this, we built the chimeric cDNA clone N605_MKHCPro, containing the HC-Pro cistron of isolate MK in the background of the N605 clone.

**Table 2. Symptoms on tobacco (Vc: vein clearing; VN: vein necrosis) and amino acid residues at critical positions of the HC-Pro of various PVY isolates.**

| Isolate/clone | Strain | Tobacco | aa196 | aa204 | aa330 | aa391 | aa392 | aa410 | Acc. No. | Reference |
|---|---|---|---|---|---|---|---|---|---|---|
| MK | NTN | Vc | D | N | N | K | T | E | MF440322 | This study |
| N605-published | N | VN | D | N | N | K | I | E | X97895 | [13] |
| N605-clone | N | VN | D | S | N | K | I | E | - | This study |
| L26 | NTN | Vc | G | N | N | K | I | E | FJ204165 | [8] |
| PVY-12 | NTN | Vc | D | N | N | K | I | E | AB185833 | [18] |
| SASA61 | N | Vc | D | N | N | K | T | E | AJ585198 | [32] |
| LW | W | Vc | D | N | N | K | I | E | AJ890349 | [33] |
| O139 | O | Vc | D | N | D | R | I | D | U09509 | [34] |

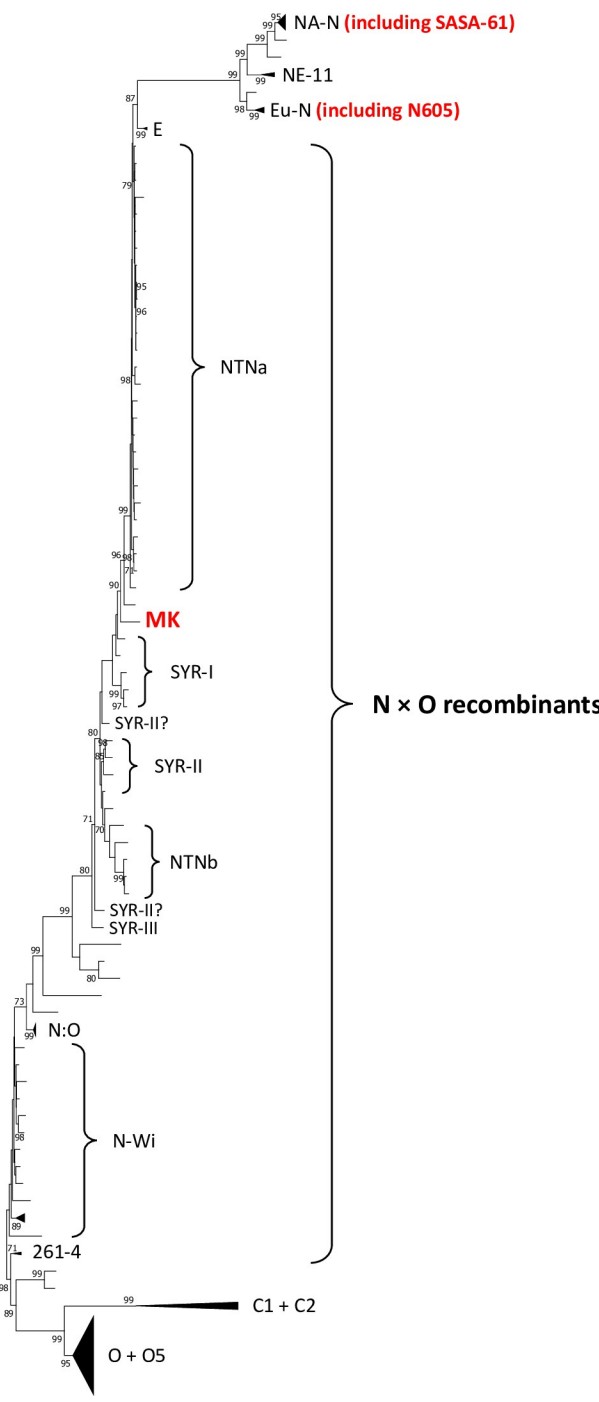

**Fig 2. Maximum-likelihood phylogenetic tree obtained with software MEGA7.0.26 of full-length genome sequences of isolate MK, isolate PVY-12 (accession number AB185833) and the 166 PVY isolates described in Table 3 in Green et al. [35].** The nucleotide substitution model GTR+G+I (General Time Reversible with Gamma distributed rate variation among Invariable sites) was selected. The consensus of 100 bootstrap replicates is shown and bootstrap values higher than 70% are indicated. Group names are according to Green et al. [35] and isolates not associated with group names are "unclassified" in Green et al. [35].

In five out of five inoculated plants of tobacco cultivar Xanthi, N605 induced veinal necrosis but N605_MKHCPro induced only mosaic and vein clearing at the systemic level (Fig 3B).

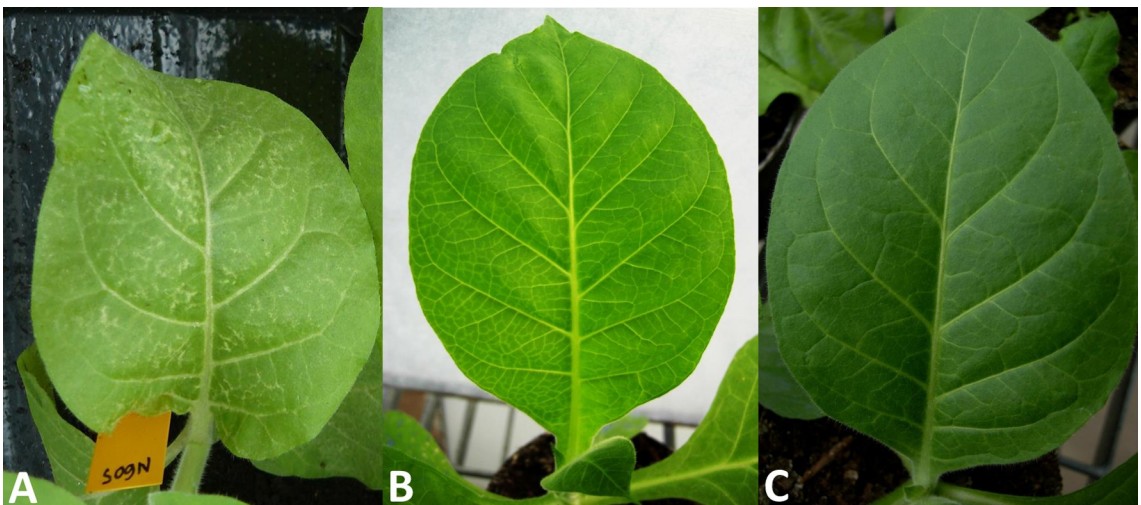

**Fig 3.** Symptoms of vein necrosis observed in tobacco cultivar Xanthi after inoculation with PVY N605 infectious clone (A) and of vein clearing after inoculation with clone N605_MKHCPro (B), compared with mock-inoculated plant (C).

### Accumulation of PVY N605 clone and chimeric clone N605_MKHCPro in tobacco

We tested if the infectious chimera N605_MKHCPro would have acquired the ability to multiply in tobacco better that the wild type PVY_N605, demonstrating a gain of fitness in such host. Results obtained in quantitative ELISA clearly stated that there was no difference in virus accumulation at 15 and 30 dpi between the wild type PVY_N605 and the chimera N605_MKHCPro (Fig 4).

## Discussion

In this study, we have identified a novel PVY determinant -involved in the triggering of systemic necrosis in tobacco. The MK isolate of PVY, originating from *D. metel*, did not induce systemic necrosis in the Xanthi tobacco cultivar despite the presence of a lysine at amino acid position 391 and a glutamic acid at amino acid position 410 of the HC-Pro, a combination of amino acids that usually determines systemic necrosis in tobacco [7]. Comparison of HC-Pro sequences of isolates MK and N605 followed by reverse genetics validated the involvement of the HC-Pro cistron of MK in the absence of necrosis in tobacco. The absence of necrosis was not due to a lower multiplication capacity of the N605_MKHCpro chimera, carrying the HC-Pro cistron of isolate MK in the background of the N605 clone, since it accumulated at levels similar to N605 in Xanthi (Fig 4). Since the only amino acid difference in the HC-Pro between the MK isolate and the N605 clone corresponds to amino acid 392 (threonine for MK and isoleucine for N605) and because all determinants of necrosis in tobacco identified so far correspond to nonsynonymous substitutions, it is highly probable that the $I_{392}T$ substitution is responsible for the absence of necrosis in tobacco plants infected by the MK isolate.

Systemic necrosis following PVY infection is responsible for considerable yield losses in tobacco. These symptoms are induced mainly by the N (Necrotic) group of PVY isolates or recombinant isolates with a HC-Pro cistron similar to that of the N group. Another PVY phylogenetic group, comprising pepper and tobacco isolates from Chile, also induces systemic veinal necrosis in tobacco [36,37]. Among tobacco cultivars that do not carry any recessive resistance gene, only a minority (10 of 162) does not exhibit necrotic symptoms upon PVY

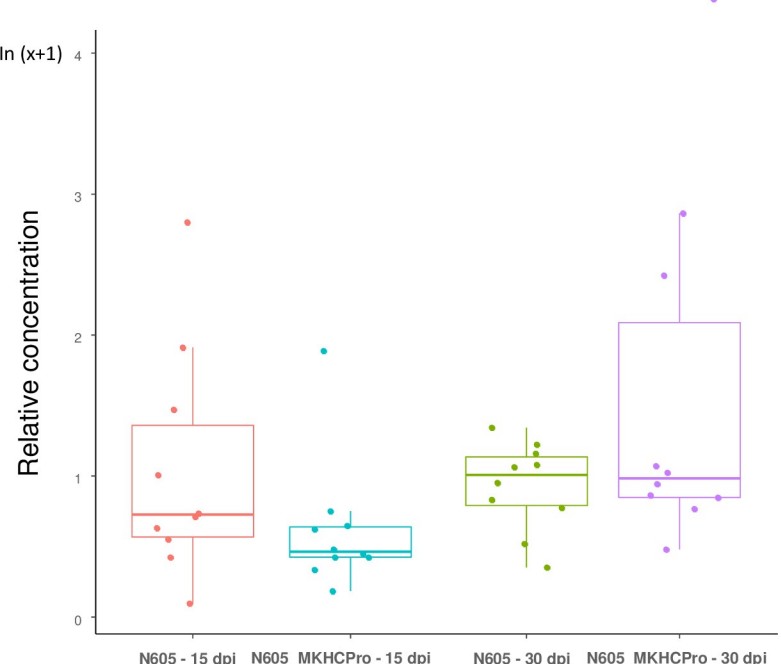

**Fig 4. Relative accumulation of N605 and the N605_MKHCPro chimera in *N. tabacum* cv. Xanthi plants at 15 and 30 days post inoculation.** Quantitative ELISA was performed on apical uninoculated leaves on independent sets of 10 plants per virus per date. A Kruskal-Wallis test [30] did not reveal any significant difference between the two viruses at each date (p-value > 0.05).

infection [38]. The *NtTPN1* gene of tobacco, a homologue of *Arabidopsis thaliana* resistance gene *RPP8* [39], determines the expression of systemic necrosis or PVY tolerance (lack of necrosis) [40]. Hence, it is highly probable that the necrotic phenotypic response induced by PVY in tobacco is similar to an induced plant resistance involving a hypersensitive response, as in the interaction between the tobacco *N* gene and tobacco mosaic virus (TMV) helicase [41], but not efficient enough to restrict rapidly the virus spread into the plant.

Two nonsynonymous substitutions at codon positions 391 and 410 in PVY HC-Pro cistron were shown to be critical in the induction of necrosis in tobacco. Indeed, mutations $K_{391}R$ or $E_{410}D$ independently abolish the ability to induce necrosis in the context of the "N" type strain N605 [7]. However, several PVY isolates described in the following encode the canonical amino acids of necrotic strains at these two positions but do not induce systemic necroses in tobacco:

- The L26 potato isolate (belonging to group NTNa; Table 2) possesses the $D_{196}G$ HC-Pro mutation that was shown to determine the absence of necrotic properties [8].

- The MK and SASA-61 isolates (closely related to the SYR-I group and belonging to group NA-N, respectively) carry the $I_{392}T$ mutation analyzed in the present article.

- The LW [33] isolate belongs to group N-Wi and no mutation responsible for the absence of necrotic properties has been identified [4]. Several amino acid mutations in the HC-Pro distinguish isolate LW from isolates of the same group that induce vein necrosis in tobacco and may be responsible for the absence of vein necrosis (mutations K431E, D256E and V91A). Alternatively, mutations in regions outside the HC-Pro may also be involved as shown in [4].

- The PVY-12 [18] isolate belongs to group NTNb and, as for the previous isolate, no mutation responsible for this phenotype has been identified. Again, this could be due mutations in the

HC-Pro cistron (PVY-12 carries mutation M264V that is absent in necrotic isolates of the same group) or elsewhere in the genome.

- Finally, several isolates of an "N:O" (or "N:O minus") group collected in the USA in 2004 or later carry mutations at position 330 of HC-Pro that may be involved in the lack of necrotic properties in tobacco [4,35].

However, some of these non-necrotic isolates do not carry any characterized candidate HC-Pro mutation that may be responsible for this phenotype. As for LW or PVY-12, the phenotype could be due to other, uncharacterized HC-Pro mutations or to mutations elsewhere in the genome.

The fact that two consecutive amino acid positions in the HC-Pro (391 and 392) are involved in tobacco necrosis suggests that this trait depends on a change in HC-Pro local surface charge and/or structure that alters its binding with a ligand in tobacco cells. However, the nature of amino acid mutations does not clearly support this hypothesis. At position 391, the lysine and arginine belong to the same group of positively-charged amino acids. At position 392, the isoleucine belongs to the hydrophobic group and the threonine to the polar group of amino acids. Therefore, no general rule can be seen on the mutations at these two positions that affect tobacco necrosis.

Phylogenetic analyses showed that the loss of necrosis in tobacco has occurred several times independently during PVY evolution. At least four independent evolutions towards a loss of

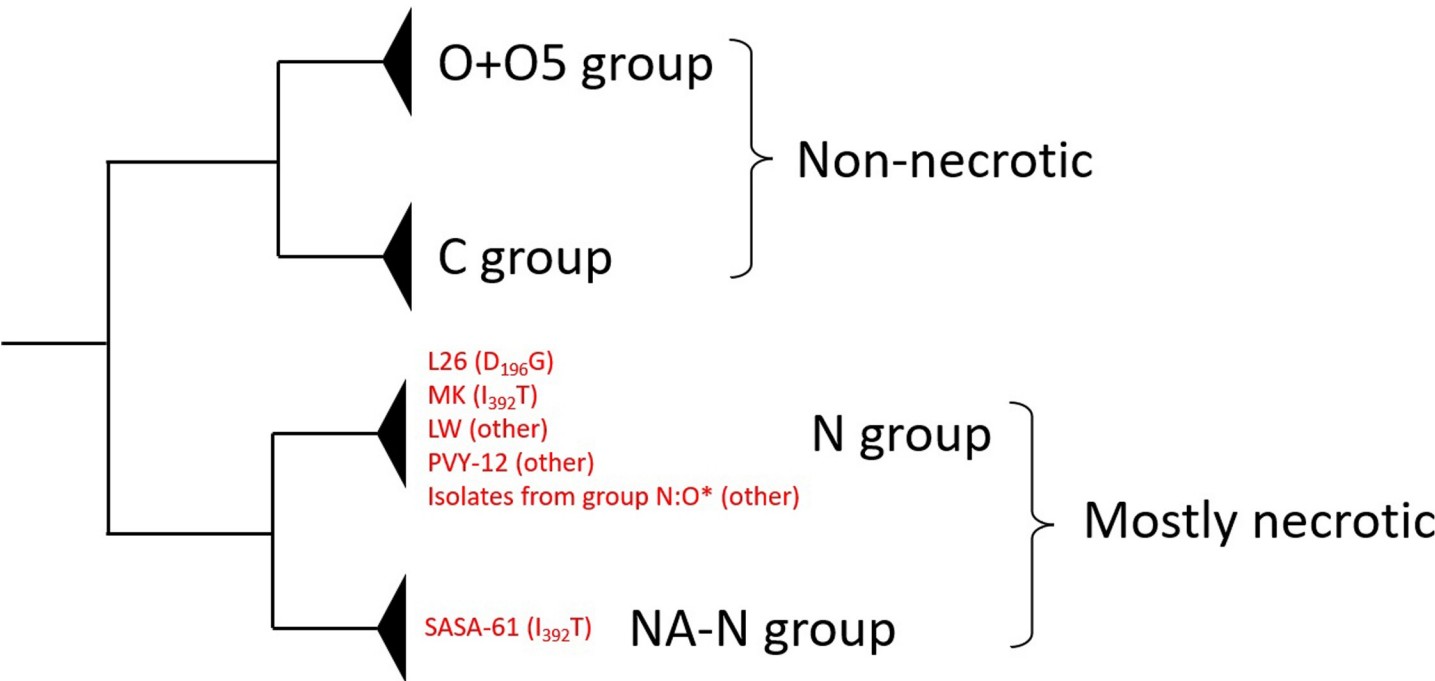

**Fig 5. Schematic phylogenetic tree of non-recombinant PVY HC-Pro coding sequences.** The tree was built with the maximum-likelihood method implemented in software MEGA7.0.26 using 157 PVY isolates described in Tab 3 in Green et al. [35] plus isolates PVY-12 (accession number AB185833) and MK (this study). The nucleotide substitution model TN92+G (Tamura-Nei model with gamma distributed rate variation among sites) was selected. The main PVY groups based on bootstrap values higher than 70% are indicated. Group names are according to Green et al. [35]. The N group includes the Eu-N group and several recombinant groups described in Green et al. [35]. Isolates carrying the canonical amino acids at positions 391 and 410 of the HC-Pro of necrotic PVY isolates but that do not induce a systemic necrosis in tobacco are indicated in red together with the mutation involved in the loss of necrosis, when identified. The asterisk correspond to isolates ME81, MI090004, MN10c_26, MN121, MN13a_39, ND121, ND68, NY51 and WI120127 from group N:O. Three of them (ME81, MN10c_26, NY51) carry mutations $N_{330}K$ or $N_{330}S$ in the HC-Pro that may be responsible for the lack of necrosis in tobacco [4].

necrotic properties in tobacco are inferred: two involving mutation $I_{392}T$ for isolates MK and SASA-61, which belong to two separate PVY clades, one involving mutation $D_{196}G$ for isolate L26 and at least one involving other, undetermined, mutations for isolates LW, PVY-12 and several other isolates in group N:O [35], for which the HC-Pro cistron sequence belongs to the same PVY clade (Fig 5).

These repeated evolutions strongly suggest that tobacco necrosis is a costly trait in PVY and that the lack of necrosis was selected for several times independently. However, the reasons for this selection are not obvious, since we did not observe any difference in PVY accumulation at the systemic level between N605 and the N605_MKHCPro chimera in Xanthi tobacco plants (Fig 4). Several scenarios can be proposed and would be interesting to test further: (i) selection of PVY mutants in tobacco cultivars other than Xanthi where the absence of necrosis induction may confer a multiplication advantage; (ii) selection of PVY mutants in other hosts for a fitness advantage independently of virulence traits, notably in certain potato cultivars carrying the *Ny* and/or *Nc* resistance genes which interact with the HC-Pro of PVY [42,43]; (iii) selection of PVY mutants for increased aphid transmissibility since HC-Pro is one of the two PVY proteins with the coat protein that are directly involved in aphid transmission.

## Supporting information

**S1 Fig. Schematic diagram of recombinant structure of the MK isolate, showing the nucleotide position intervals for the three putative breakpoints identified by seven out of nine methods implemented in the RDP2 package with a 99% of confidence interval [25].** The organization of the PVY genome (ca. 9.7-kb long) is reported on the top, where individual cistrons are presented as rectangles with corresponding protein names. The putative parental sequences of PVY-MK, i.e. PVY group O in white and PVY group Eu-N in black, are shown below the PVY genome. The recombinant structure of group SYR-I [13] is also reported to show its similarity with that of PVY-MK.
(TIF)

**S1 Table. Primers used for genome amplification and sequencing of PVY-MK.**
(DOCX)

## Acknowledgments

This work is dedicated to Prof. Crisostomo Vovlas ("Makis" for us) from the University "Aldo Moro" of Bari (Italy), who gave us the symptomatic *Datura metel* plant analyzed in this study. We would also like to thank Mr. Salvatore Cristadoro and Mrs. Immacolata Nunziata (IPSP-CNR) for their administrative support.

## Author Contributions

**Conceptualization:** Giuseppe Parrella, Benoit Moury.

**Data curation:** Giuseppe Parrella, Benoit Moury.

**Formal analysis:** Giuseppe Parrella, Benoit Moury.

**Funding acquisition:** Giuseppe Parrella, Benoit Moury.

**Investigation:** Giuseppe Parrella, Benoit Moury.

**Methodology:** Giuseppe Parrella, Benoit Moury.

**Project administration:** Giuseppe Parrella, Benoit Moury.

**Resources:** Giuseppe Parrella, Benoit Moury.

**Software:** Giuseppe Parrella, Benoit Moury.

**Supervision:** Giuseppe Parrella, Benoit Moury.

**Validation:** Giuseppe Parrella, Benoit Moury.

**Visualization:** Giuseppe Parrella, Benoit Moury.

**Writing – original draft:** Giuseppe Parrella, Benoit Moury.

**Writing – review & editing:** Benoit Moury.

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
