## [Decision Letter · Decision Letter 0]

1 Apr 2024

PONE-D-24-09166A new point mutation in the HC-Pro of potato virus Y is involved in tobacco vein necrosisPLOS ONE

Dear Dr. Parrella,

Thank you for submitting your manuscript to PLOS ONE. After careful consideration, we feel that it has merit but does not fully meet PLOS ONE’s publication criteria as it currently stands. Therefore, we invite you to submit a revised version of the manuscript that addresses the points raised during the review process.

**ACADEMIC EDITOR: Address all the comments of the peer-reviewers and resubmit your manuscript for further consideration. **

We look forward to receiving your revised manuscript.

Kind regards,

S.V. Ramesh, PhD

Academic Editor

PLOS ONE

Journal Requirements:

   "This research was partially funded by the Italian National Research Council (CNR) with the Short Term Mobility (STM) program 2014."

Additional Editor Comments:

Peer-reviewers see potential in your work nevertheless pending some revisions in the manuscript.

Reviewers' comments:

Reviewer's Responses to Questions

**Comments to the Author**

1. Is the manuscript technically sound, and do the data support the conclusions?

Reviewer #1: Yes

Reviewer #2: Yes

2. Has the statistical analysis been performed appropriately and rigorously? 

Reviewer #1: Yes

Reviewer #2: Yes

3. Have the authors made all data underlying the findings in their manuscript fully available?

Reviewer #1: Yes

Reviewer #2: Yes

4. Is the manuscript presented in an intelligible fashion and written in standard English?

Reviewer #1: Yes

Reviewer #2: Yes

5. Review Comments to the Author

Reviewer #1: The manuscript entitled “A new point mutation in the HC-pro of potato virus Y is involved in tobacco vein necrosis” described a new determinant of TVN in the MK isolate of PVY, though it did not induce systemic necrosis in tobacco. The subject is interesting, and the work has been done thoroughly.

The manuscript is well-structured overall, but it requires meticulous editing with a focus on English grammar, spelling, and sentence structure. I think this manuscript may be published in “PLOS ONE” Journal.

Some minor comments should be addressed before acceptance:

Page 1, Line 26: Virus names should be written in non-italics.

In the Introduction, some sentences begin inappropriately and need to be rephrased. For example,

Page 1, Lines 30-36: The word "PVY" was consistently used at the start of each sentence. Please edit the sentences to eliminate the need for repeating "PVY" at the beginning of each sentence.

In introduction, lines 54-56: This sentence is too long. The expressions and English need to be improved.

Lines 114-115: This sentence needs to be revised.

Lines 117-119: What do the authors mean by “for all programs implemented”?

Did the authors consider using Burt to verify the break point positions?

Moreover, please elucidate why the Burt algorithm was not used to verify the recombination?

The resolution and quality of all figures are not satisfactory. The authors are advised to improve them for a better presentation.

Line 226: Due to the very low resolution and quality of the phylogenetic tree figures, it is unclear whether the authors are referring to a “Branch” or a “Clade”.

I strongly recommend authors to improve the visuals of Figures 2 and 4 for better clarity, as nothing is clear to me.

Reviewer #2: The manuscript identified a new PVYN-like amino acid in PVY HCPro associated with the induction of veinal necrosis (VN) in tobacco (I392 induction of VN, T392 loss of VN). The PVY isolate MK naturally occurred in a plant of D. metel. MK genomic sequence aligned with PVYNTN group, serologically it is a PVYN serotype, and biologically it unexpected doesn’t induce VN in tobacco. An infectious clone with the replaced MK-HCPro in N605 isolate confirmed the loss of VN character. A single amino acid polymorphism (SAP) at HCPro position 392 was found responsible for the loss of VN in MK, which is an important finding in PVY study. However, the manuscript needs minor revision.

Some detailed comments are indicated below:

In Abstract, should point out which amino acid in position 392 (should be T392) is related to the loss of VN.

In Abstract line 17-18, in fact it is not the HCPro protein was replaced, but the sequence coding for HCPro protein was replaced.

Line 26, about virus name, to my knowledge, the combination of virus name in italic and abbreviation is not proper, based on the standard at: https://ictv.global/faq/names:

“A species name* is written in italics with the first word beginning with a capital letter. Other words only begin with a capital if they are proper nouns (including host genus names but not virus genus names**) or alphabetical identifiers. A species name should not be abbreviated.”

Line 44, about the strain naming, PVYW? In general, the manuscript should follow the same standard for naming the strains, indicate strain naming was based on which publication(s).

Line 55, “molecular characteristics” – lines 55-56 only explained the unusual biological character relating to the serotype. How unusual molecularly is not explained in this place, or it is not known before study by this manuscript.

Line 72, what kind of RT-PCR? For strain typing or just for PVY positive or negative assessment?

Line 76, should explain what is a PVYOb strain?

The same for Line 91 PVYOa (what is a PVTOa strain?)

Line 101: if possible, explain why to choose PVY-12 as the primers based on? MK has the same RT-PCR pattern as PVY-12?

Line 143-144: should explain briefly how the “relative PVY concentration” was counted based on? Relative to what?

The same for figure legend of Line 258, Fig. 4: explain briefly “how relative accumulation was estimated”

Line 232: if possible, explain briefly “nucleotide substitution model GRT+G+I”.

The same for Line 337, explain briefly “nucleotide substitution model TN92+G”.

6. PLOS authors have the option to publish the peer review history of their article (what does this mean?). If published, this will include your full peer review and any attached files.

Reviewer #1: No

Reviewer #2: No

---

## [Author Response · Author response to Decision Letter 0]

6 Apr 2024

RESPONSES TO REFEREES

Dear Editor,

we wish to thank you and the three referees which have made an important improvement of the paper with their suggestions/comments. We have prepared a revised version of the manuscript following all the suggestions of the referees and below we have reported our responses point by point (responses are in red).

Reviewer #1:

1) Page 1, Line 26: Virus names should be written in non-italics. 

R: Virus name corrected.

2) In the Introduction, some sentences begin inappropriately and need to be rephrased. For example,

Page 1, Lines 30-36: The word "PVY" was consistently used at the start of each sentence. Please edit the sentences to eliminate the need for repeating "PVY" at the beginning of each sentence.

R: Sentences were rephrased accordingly to referee suggestion.

3) In introduction, lines 54-56: This sentence is too long. The expressions and English need to be improved.

R: Sentences were made shorter and were rephrased with the aim to improve English following referee suggestion.

4) Lines 114-115: This sentence needs to be revised.

R: The sentence was revised and rephrased.

5) Lines 117-119: What do the authors mean by “for all programs implemented”?

R: the sentence was rephrased in order to clarify the concept.

6) Did the authors consider using Burt to verify the break point positions?

R: we used RDP2 which do not include the Burt method.

7) Moreover, please elucidate why the Burt algorithm was not used to verify the recombination?

R: please see the previous response.

8) The resolution and quality of all figures are not satisfactory. The authors are advised to improve them for a better presentation.

R: we checked all the figures with PACE tool as recommended and the uploaded figured have passed the PLOS requirements.

9) Line 226: Due to the very low resolution and quality of the phylogenetic tree figures, it is unclear whether the authors are referring to a “Branch” or a “Clade”.

R: the quality of figures have been improved following PLOS requirements.

10) I strongly recommend authors to improve the visuals of Figures 2 and 4 for better clarity, as nothing is clear to me.

R: please see the previous responses.

Reviewer #2: 

1) In Abstract, should point out which amino acid in position 392 (should be T392) is related to the loss of VN.

R: In the Abstract it has been indicated the amino acid putatively related to the loss of VN.

2) In Abstract line 17-18, in fact it is not the HCPro protein was replaced, but the sequence coding for HCPro protein was replaced.

R: the sentence has been rephrased following subggestion.

3) Line 26, about virus name, to my knowledge, the combination of virus name in italic and abbreviation is not proper, based on the standard at: https://ictv.global/faq/names:

“A species name* is written in italics with the first word beginning with a capital letter. Other words only begin with a capital if they are proper nouns (including host genus names but not virus genus names**) or alphabetical identifiers. A species name should not be abbreviated.”

R: the virus name has been corrected following ICTV rules.

4) Line 44, about the strain naming, PVYW? In general, the manuscript should follow the same standard for naming the strains, indicate strain naming was based on which publication(s).

R: The PVYW is a recombinant of PVYN named “Wilga”: the paper describing the PVY recombinants has been added in the text and references.

5) Line 55, “molecular characteristics” – lines 55-56 only explained the unusual biological character relating to the serotype. How unusual molecularly is not explained in this place, or it is not known before study by this manuscript.

R: the sentence was corrected following the suggestion.

6) Line 72, what kind of RT-PCR? For strain typing or just for PVY positive or negative assessment?

R: common RT-PCR was used following by Sanger sequencing to verify the presence over the time of the mutation in the HCPro. The approach used has been better specified in the text.

7) Line 76, should explain what is a PVYOb strain?

The same for Line 91 PVYOa (what is a PVTOa strain?)

R: The origin and the significance of the PVYOa and PVYOb has been indicated in the text by adding a specific reference.

8) Line 101: if possible, explain why to choose PVY-12 as the primers based on? MK has the same RT-PCR pattern as PVY-12?

R: We chose PVY-12 isolate because it showed the same biological and serological characteristics as MK isolate.

9) Line 143-144: should explain briefly how the “relative PVY concentration” was counted based on? Relative to what?

R: We speak about the “relative concentration” of the virus because in the analysis, done on different days, the viral concentration is correlated with a serial dilution (which is used to construct a regression line), using an extract from an infected plant of which the true concentration of the virus, expressed in units of weight, is unknown. This approach is frequently used in experiments with the aim to estimate virus accumulation in plants over time by using absorbance in quantitative ELISA. It is called "relative" because it refers to absorbance values that are related to the virus concentration but does not express the absolute values of viral concentration, which are generally expressed as weight/volume or weight/weight.

10) The same for figure legend of Line 258, Fig. 4: explain briefly “how relative accumulation was estimated”

R: please see the previous response.

11) Line 232: if possible, explain briefly “nucleotide substitution model GRT+G+I”.

The same for Line 337, explain briefly “nucleotide substitution model TN92+G”.

R: a brief explanation of the significance of the two substitution models has been added in the text.

---

## [Editor Report · Decision Letter 1]

9 Apr 2024

A new point mutation in the HC-Pro of potato virus Y is involved in tobacco vein necrosis

PONE-D-24-09166R1

Dear Dr. Parrella,

We’re pleased to inform you that your manuscript has been judged scientifically suitable for publication and will be formally accepted for publication once it meets all outstanding technical requirements.

Kind regards,

Shunmugiah Veluchamy Ramesh, PhD

Academic Editor

PLOS ONE
---

## [Editor Report · Acceptance letter]

26 Apr 2024

PONE-D-24-09166R1 

PLOS ONE

Dear Dr. Parrella, 

I'm pleased to inform you that your manuscript has been deemed suitable for publication in PLOS ONE. Congratulations! Your manuscript is now being handed over to our production team.

Kind regards, 

on behalf of

Dr. Shunmugiah Veluchamy Ramesh 

Academic Editor

PLOS ONE